# CytokineLink: A Cytokine Communication Map to Analyse Immune Responses—Case Studies in Inflammatory Bowel Disease and COVID-19

**DOI:** 10.3390/cells10092242

**Published:** 2021-08-29

**Authors:** Marton Olbei, John P. Thomas, Isabelle Hautefort, Agatha Treveil, Balazs Bohar, Matthew Madgwick, Lejla Gul, Luca Csabai, Dezso Modos, Tamas Korcsmaros

**Affiliations:** 1Earlham Institute, Norwich NR4 7UZ, UK; marton.olbei@earlham.ac.uk (M.O.); drjohnpthomas@gmail.com (J.P.T.); isabelle.hautefort@earlham.ac.uk (I.H.); agatha.treveil@gmail.com (A.T.); balazs.bohar@earlham.ac.uk (B.B.); matthew.madgwick@earlham.ac.uk (M.M.); lejla.gul@earlham.ac.uk (L.G.); luca.csabai@earlham.ac.uk (L.C.); dezso.modos@quadram.ac.uk (D.M.); 2Quadram Institute Bioscience, Norwich NR4 7UZ, UK; 3Department of Gastroenterology, Norfolk and Norwich University Hospital, Norwich NR4 7UZ, UK; 4Department of Genetics, Eotvos Lorand University, 1117 Budapest, Hungary

**Keywords:** cytokine, network, resource, Cytoscape, NDEx, COVID-19, inflammatory bowel disease

## Abstract

Intercellular communication mediated by cytokines is critical to the development of immune responses, particularly in the context of infectious and inflammatory diseases. By releasing these small molecular weight peptides, the source cells can influence numerous intracellular processes in the target cells, including the secretion of other cytokines downstream. However, there are no readily available bioinformatic resources that can model cytokine–cytokine interactions. In this effort, we built a communication map between major tissues and blood cells that reveals how cytokine-mediated intercellular networks form during homeostatic conditions. We collated the most prevalent cytokines from the literature and assigned the proteins and their corresponding receptors to source tissue and blood cell types based on enriched consensus RNA-Seq data from the Human Protein Atlas database. To assign more confidence to the interactions, we integrated the literature information on cell–cytokine interactions from two systems of immunology databases, immuneXpresso and ImmunoGlobe. From the collated information, we defined two metanetworks: a cell–cell communication network connected by cytokines; and a cytokine–cytokine interaction network depicting the potential ways in which cytokines can affect the activity of each other. Using expression data from disease states, we then applied this resource to reveal perturbations in cytokine-mediated intercellular signalling in inflammatory and infectious diseases (ulcerative colitis and COVID-19, respectively). For ulcerative colitis, with CytokineLink, we demonstrated a significant rewiring of cytokine-mediated intercellular communication between non-inflamed and inflamed colonic tissues. For COVID-19, we were able to identify cell types and cytokine interactions following SARS-CoV-2 infection, highlighting important cytokine interactions that might contribute to severe illness in a subgroup of patients. Such findings have the potential to inform the development of novel, cytokine-targeted therapeutic strategies. CytokineLink is freely available for the scientific community through the NDEx platform and the project github repository.

## 1. Introduction

Cytokines are the key mediators of intercellular communication. Cytokines are low molecular weight peptides (5–20 kDa) released by a wide repertoire of cell types including immune cells, epithelial cells, endothelial cells, fibroblasts, and other stromal cells. They enable coordinated autocrine, paracrine, and even endocrine homeostatic functions to occur within the body [1]. Cytokines exert these effects by binding to specific receptors that are expressed by target cells, which activate downstream intracellular signalling pathways and ultimately result in the expression of genes required for a particular process [2]. Cytokines effectively regulate all physiological processes in the human body ranging from embryonic development, to angiogenesis, to haemostasis, and innate and adaptive immunity. Cytokines can achieve such diverse and wide-ranging effects because they exhibit pleiotropy (i.e., a single cytokine can cause different biological effects and can act on different cell types) and redundancy (i.e., a variety of cytokines can activate the same signalling pathways) [3].

In addition, cytokine signalling is highly fine-tuned and regulated with cytokines themselves affecting the activity of each other through synergetic, antagonistic, or additive effects and feedback loops [4]. However, disturbances in the regulation of cytokine communication can contribute to pathological processes including cancers, autoimmune disorders (e.g., inflammatory bowel disease (IBD)), and cytokine storms, as seen in response to certain infections (e.g., SARS-CoV-2) [5,6,7]. Although great progress has been made in identifying, characterising, and understanding the targets and functions of cytokines in the human body in health and disease, there is a lack of readily available resources that can computationally model cytokine–cytokine interactions [8,9]. Hence, we developed a novel biological network resource, CytokineLink. CytokineLink includes 10313 total interactions from 18 healthy human cell subsets and 24 tissue types. With CytokineLink two types of indirect or metanetworks were defined: (1) a cell–cell communication network connected by cytokines, and (2) a cytokine–cytokine interaction network depicting the potential ways in which cytokines can affect the activity of each other. To illustrate the utility and real-world applicability of this novel resource, we have demonstrated how CytokineLink can reveal important insights into cytokine signalling in IBD and following SARS-CoV-2 infection.

## 2. Materials and Methods

### 2.1. Network Generation

We built the CytokineLink network resource based on cytokine–receptor interactions found in the literature [1] and the integrated OmniPath resource [10]. To achieve broad coverage of cytokines, all cytokines listed in the ImmuneXpresso and ImmunoGlobe systems of immunology databases were selected. To define the tissue- or cell-type-specific expression properties for all these cytokines and their receptors, we downloaded the consensus RNA-Seq data from the Human Protein Atlas (HPA) website [11,12]. We used the enrichment of consensus expression from the downloaded HPA data as designated by HPA. HPA was selected as the source of expression data due to its expansive coverage, both in terms of the queried cytokines and receptors, and the available expression data from blood cells and tissues. The cytokine–receptor interactions, and the tissue/cell associations of all cytokines and cytokine receptors were merged into an integrated edge list. This edge list enabled us to reconstruct a base network describing the chain of cytokine signalling events occurring between a source cell and target tissue types, connected by the included cytokine–cytokine–receptor interactions.

From the reconstructed base network, we created two metanetworks. The metanetworks were created by generating all the shortest paths of the length of five in the base network, with the all_simple_paths() function of the NetworkX (version: 2.5) python library (python version: 3.8.6). The resulting list of shortest paths was then filtered to contain only the paths capturing the adequate chain of events (i.e., Cell1, Cytokine1, Receptor, Cell2, Cytokine2). These paths, containing the underlying signalling events, were used to generate the cell–cell and cytokine–cytokine meta-interactions, as seen in Figure 1.

### 2.2. Annotation

The systems of immunology databases, immuneXpresso and ImmunoGlobe, curate interaction data collated from the literature, between cell types and cytokines, both as sources and targets (i.e., both the cytokines and cells can be actors and can be acted upon). The curated cell–cytokine and cytokine–cell interaction annotations from immuneXpresso and ImmunoGlobe were included in CytokineLink, assigning the literature enrichment to the cell–cytokine associations to the underlying edges in the general network the meta-edges were created from. In practice, following the example in Figure 1, this means that for every meta-edge (e.g., myeloid DC → memory B-cell) users can see if the cytokine–cell interaction annotations establishing the meta-edge (i.e., myeloid DC → TNFSF13) has been reported in the literature previously. Appendix A contains the combined cytokine–cell interaction annotations from the ImmuneXpresso and ImmunoGlobe databases.

### 2.3. Assessing the Translational Relevance of the Cytokine–Cytokine Interactions

To measure the relevance of cytokine–cytokine interactions, we compared the differentially expressed genes signatures from experiments where patients were treated with cytokine inhibitor drugs. The effects of infliximab, golimumab (two TNFα inhibitors), and tocilizumab (an IL6 inhibitor) were investigated. The significance of differentially expressed cytokines regulated by TNFα or IL6 following inhibitor treatment was determined using a chi-square test of independence, via the chisq.test() R language function. Expression data were downloaded from GEO and analysed using GEO2R. The cutoff for differential expression was a fold-change of 2, with an adjusted *p*-value of less than 0.05. The R scripts used for establishing differential gene expression and the subsequent statistical analysis can be found in our GitHub repository: https://github.com/korcsmarosgroup/CytokineLink (accessed on 2 August 2021). Table 1 lists, and briefly describes, the datasets we used.

### 2.4. Use Cases

For the evaluation of cytokine communication in ulcerative colitis (UC), single-cell expression data from patients containing 51 colonic cell subsets were used and processed from the seminal study by Smillie et al. [15]. All genes with an RNA count of 0 were removed, and normalised log2 counts across all samples (separately for each cell type) were fitted to a Gaussian kernel [16]. Only the genes falling into the top quintile of expression ranges were considered expressed, both for cytokines and receptors, respectively. From the processed data, 13 cell subsets that are especially important in inflamed ulcerative colitis (out of 51 cell subsets) were selected [17,18,19]. The cell type names were used as presented in Smillie et al. [15]: inflammatory fibroblasts, myofibroblasts, goblet cells, immature goblet cells, M cells, CD8+ IEL, CD8+ LP, CD8+ IL17+, DC1, DC2, inflammatory monocytes, and Tregs.

To evaluate the applicability of the tool for infectious diseases, we used cytokines elevated in COVID-19 patients collected for our previous work [20]. The elevated cytokine list contained: CCL2, CCL4, CSF2, CXCL10, CXCL8, IL6, IFNG, IL1B. A complemented cytokine list containing cytokines elevated in viral infections capable of causing cytokine release syndrome (CRS), but not elevated in COVID-19, contained IL4 and IL5, also from our previous work [20].

## 3. Results

### 3.1. A Novel Network Resource of Cytokine Communication

To deconvolute the effects of cytokines on intercellular communication, we built a new network resource that aimed to depict all possible indirect interactions (i.e., regulatory interactions) of the most prevalent cytokines from the literature and overlaid it using consensus expression data from the HPA [21]. CytokineLink was built using 18 blood cell types and 24 tissues, using 115 cytokines with 260 cytokine–receptor pairs. To add more confidence to the interactions, annotation data from two systems of immunology databases (immuneXpresso, and ImmunoGlobe) were added that contains the literature enrichment scores, signage (i.e., stimulatory/inhibitory) data, and the literature references for interactions [11,12]. Table 2 lists the number of interactions and degree of the literature annotation in CytokineLink. The annotated interactions section in the table refers to at least one of the two cell–cytokine interactions in the chain of signalling events (i.e., Cell1–Cytokine1–Receptor–Cell2–Cytokine2) being listed in the literature as per the ImmunoGlobe and immuneXpresso databases.

### 3.2. Cytokine–Cytokine Meta-Edges Capture the Possible Ways Cytokines Affect Each Other

To capture the validity of cytokine–cytokine meta-edges, we analysed publicly available gene expression datasets of cytokine inhibitors (infliximab, golimumab, and tocilizumab) for two important proinflammatory cytokines, TNFα and IL6. The chosen gene expression datasets measured the abundance of RNA in patients suffering from chronic illnesses where cytokine communication is often disrupted (inflammatory bowel disease, rheumatoid arthritis) [22,23]. Analysing the differential expression between the treatment and control groups, we investigated whether there is a significant association between the differential expression of cytokines in the selected datasets, and the cytokines captured by our model. To summarise, we tested if there was a significant association in differentially expressed cytokines between the targets of one of the inhibited cytokines (e.g., IL6), and differentially expressed cytokines that were not targeted by the inhibited cytokine in CytokineLink. We were able to confirm a significant relationship for IL6 and TNFα using a chi-square test of independence, as we found a statistically significant difference between the presence of differentially expressed cytokines that were not targeted by the respective cytokines, compared to the differentially expressed cytokines that were targets of IL6 (tocilizumab *p* value = 0.00587), and TNFα (infliximab *p* value = 0.01504, golimumab *p* value = 0.01525) in our model. These results highlight that CytokineLink can capture physiologically relevant cytokine–cytokine interactions for two important proinflammatory cytokines. For detailed steps and scripts used please refer to the Methods section.

### 3.3. Cytokine Signalling in Ulcerative Colitis (UC)

Ulcerative colitis (UC) is one of the two main clinical subtypes of IBD, which is characterised by chronic inflammation of the intestinal mucosa that begins in the rectum and extends proximally along the colon [24]. It is a relapsing and remitting condition that can cause significant morbidity in patients. The incidence, as well as prevalence, of UC has been increasing globally over recent decades [24,25]. Although the aetiology of the disease is unclear, complex interactions between multiple genetic risk loci, environmental factors, and the gut microbiota are thought to give rise to dysregulated immune responses in the colon, resulting in chronic inflammation [18]. Whilst UC can be successfully treated through the surgical removal of the colon (i.e., colectomy), many patients wish to preserve the function of their colons and avoid the complications of surgery, and hence, opt for medical management which aims to induce and maintain remission [26,27].

The advent of biologic therapies targeting cytokine signalling (e.g., Infliximab targeting TNFα or, more recently, ustekinumab targeting the p40 subunit of IL12 and IL23) has transformed the medical management of UC in recent years and has led to fewer patients undergoing colectomies and better treatment outcomes [17,28,29,30]. Nevertheless, approximately 20% of UC patients fail to respond to medical therapies and eventually undergo surgical intervention for their disease [17]. One important reason for this is that certain subgroups of UC patients may possess distinct pathogenic pathways and cytokine signalling networks that are not targeted by the existing armamentarium of biologic therapies [31]. Thus, further disentangling the cytokine signalling networks driving the pathomechanisms in UC patients is a pressing need. Over the past three years, a number of landmark single-cell RNA sequencing (scRNAseq) studies have been performed in IBD patients, that have revealed critical and novel insights into the likely cell types mediating the pathogenic processes involved during non-inflamed and inflamed states of the disease [15,32,33,34]. However, these studies did not fully interrogate the cytokine-specific signalling networks occurring between these cell types. This would be important for better understanding the dysregulation of cytokine signalling in UC and revealing potential new therapeutic targets for patients.

Therefore, we applied CytokineLink to a recently published scRNAseq dataset obtained from inflamed and non-inflamed tissues of UC patients [15]. This analysis revealed novel insights into the differences in cytokine signalling between non-inflamed and inflamed UC conditions. We first mapped cell–cell interactions between 13 cell types in both inflamed and non-inflamed UC conditions (Figure 2a). We then filtered these interactions by individual cytokines, which led to the emergence of disease status-specific differences in the patterns of intercellular cytokine communication (Figure 2b). This faceted visualization approach can highlight cytokine-specific signalling differences occurring between disease states and cell types. For example, the generally anti-inflammatory cytokine IL10 was found to be produced by regulatory T cells (Treg cells) in both conditions, but in non-inflamed tissues additional interactions arose involving innate lymphoid cells (ILCs) and type 2 dendritic cells (DC2s), stemming from an increase in the expression of the IL10 receptor (IL10R) in these cell types. The difference in interactions between disease states was even more apparent in the case of TGFB1, where there was only a small overlap between TGFB1-mediated intercellular signalling in both inflamed and non-inflamed UC tissues. In non-inflamed UC conditions, TGFB1 was found to drive multiple interactions involving CD8+ intraepithelial cells and a number of cell types including inflammatory fibroblasts, myofibroblasts, inflammatory monocytes, and type 2 dendritic cells (DC2s). However, in inflamed UC conditions, TGFB1 signalling shifted towards interactions occurring largely between type 1 dendritic cell cells (DC1s) and inflammatory fibroblasts, myofibroblasts, DC2s, and CD8 lamina propria (LP) T cells.

### 3.4. Cytokine Signalling following SARS-CoV-2 Infection

During the COVID-19 pandemic, it became apparent that the development of a cytokine storm can lead to increased mortality in a subset of patients infected by SARS-CoV-2 [35]. The cytokine storm is caused by the overactivation of adaptive and innate immune cells, which can lead to a positive feedback loop of pro-inflammatory cytokines [36]. To further study pro-inflammatory cytokine signalling occurring between various cell types in COVID-19, we applied CytokineLink to a literature curation dataset containing cytokines that are upregulated following SARS-CoV-2 infection in patients [20]. Using this metanetwork, we uncovered important connections in the cytokine communication of COVID-19 patients.

We found that the majority of cell types included in CytokineLink are activated by the elevated cytokines during SARS-CoV-2 infection (Figure 3a). However, in our previous work, we noted that the levels of certain cytokines were not always elevated in COVID-19, compared to infections by other cytokine storm-causing viruses (SARS-CoV, MERS-CoV, H5N1, H7N9) [20].

Figure 3b highlights the differences in metanetworks caused by the compared cytokine profiles. This approach lets users study the system’s level changes inflicted by the addition of just a few cytokines. In this instance, we complemented the system with cytokines that are found induced in other virus-induced cytokine storms (i.e., IL4 and IL5) to show what parts of the intercellular and cytokine–cytokine communication network were affected based on literature curated data. The complemented model further increases the connectivity of basophils in the network by connecting them to multiple members of the innate immune system and connects memory B-cells to eosinophils and basophils.

On the level of cytokine–cytokine interactions the difference is more apparent, as IL5 and especially IL4 act as hubs, further increasing the connections between the elevated cytokines. The CytokineLink analysis provides a possible mechanistic link to the lack of one of the key anti-inflammatory cytokines, IL12. The low levels of IL12 are comparable to healthy controls in severe COVID-19 patients, while the levels of IL4 are significantly higher in severe patients [37,38]. We note that the basis of using IL4 to complement the system was based on our previous work involving the literature curation. At the time one research article was available that had measured serum IL4 levels in SARS-CoV-2-infected patients. These patients had mild symptoms and, therefore, low levels of IL4 [39]. Comparing the SARS-CoV-2-specific and -complemented network models, there is a missing interaction between basophils and naive B-cells, mediated by the cytokine IL4. IL4 is capable of repressing Th1 responses, including the production of IL12 and IFN-γ [40]. Thus, the increase in the levels of IL4 in severe COVID-19 patients can offer an explanation for the low levels of observed IL12, through the previously described IL4–IL12 interaction captured by our model. IL12 is an important stimulator of IFN-γ and the type II interferon response. This response is critical for leading the innate immune response into the adaptive one and eliminating viral infection. It also has an important role in facilitating the differentiation of naive B cells into plasma cells. In our analysis, we found that IL12 levels remain low in patients with severe COVID-19 likely due to an increase in IL4. IL4 is a potent inhibitor of Th1 responses, including the production of IL12 and IFN-γ. Thus, the diminished IL12 levels in a certain subgroup of COVID-19 patients may explain the reason why they experience more severe disease. Further work is needed to investigate this potential causal link [41,42,43,44].

## 4. Discussion

In this study, we have introduced a new biological network resource, CytokineLink, enabling the detailed analysis of intercellular communication mediated by cytokines. We integrated data from various interaction and expression databases and used the available literature to strengthen and demonstrate the reliability of the interactions in CytokineLink. CytokineLink is freely available for the scientific community on the NDEx platform [45].

The CytokineLink database stores cytokine interactions in two ways. The intercellular interaction meta-edges convey the interactions of the individual cell types and tissues included in the analysis, mediated by enriched cytokines. Cytokine–cytokine meta-edges indicate the potential pathways through which cytokines can affect the production of other cytokines. These two meta-interaction layers place the underlying chain of events in two different contexts using the cell–cell and cytokine–cytokine meta-edges, elucidating as to how cytokines are involved in immune responses following infection or inflammatory diseases. These metanetworks are a context-specific interpretation of the underlying signalling events, depicting the flow of information mediated by intercellular cytokine communication.

The chain of events consisting of the binding and release of cytokines between various cell types of the body have been well documented in the scientific literature [46,47]. In recent years, interest has grown in the study of intercellular interactions occurring between immune cell subtypes [15,48,49,50]. However, these studies had a broader selection of ligand–receptor interactions and used differential expression and/or marker genes describing the cell subsets to generate their interactions. CytokineLink provides metanetworks built from healthy human consensus expression data, enriched with the literature annotation information previously present in separate, unconnected resources. Our methods use the underlying chain of signalling events to generate the metanetworks, and because of this every meta-edge can be led back to the exact signalling sequence it was created from, the information content of the generated meta-edges matches those of the original signalling events.

As such, the interactions in CytokineLink represent a novel, context-specific signalling format, allowing for focused analyses involving cell–cell or cytokine–cytokine interactions. Our analyses using the CytokineLink platform highlight the rewiring of cell–cell communication between disease states in a cytokine-centered way, mediated by specific cytokines in UC, a form of IBD, and COVID-19. In this analysis, we found that a certain subgroup of patients may develop severe SARS-CoV-2 infection due to low levels of IL12 (a potent cytokine for stimulating antiviral immune responses) as a result of increased IL4 levels. This suggests that targeting IL4 may have beneficial effects in severe COVID-19 infection. In UC, the generally anti-inflammatory cytokine, IL10, was involved in intercellular communication in both non-inflamed and inflamed conditions. However, CytokineLink revealed that IL10-mediated interactions between Tregs and ILCs and DC2s, that occurred in non-inflamed conditions became absent in inflamed conditions due to the loss of IL10R expression in ILCs and DC2s. Thus, therapeutic strategies that may aim to increase IL10 levels in the colonic mucosa during inflamed UC to ameliorate inflammation, may be ineffective due to a lack of expression of the IL10R in key immune cell types. Similar analyses involving multi-omics data could help researchers shortlist potential therapeutic targets, through the integrated analysis of the expression, proteomic, or regulatory status of tracked cytokines [15,33,51]. For instance, CytokineLink could be applied to gene expression and proteomic data from patient cohorts receiving particular biologic therapies. Analysis of the resultant metanetworks will reveal cytokines or cell types that act as hubs in intercellular communication networks associated with response and non-response to biologic therapies. Cytokines acting as hubs in networks associated with non-response to biologic therapies may act as novel drug targets to induce remission in these patients.

The introduction of cytokine–cytokine interactions provides a novel meta-edge for the functional study of signalling between the elements of the immune system. Due to the limited amount of available experimental data on cytokine inhibitors we were only able to confirm a significant relationship for a small subset of cytokines, but future studies could further establish the reliability of such interactions, especially using more refined modelling steps, involving intracellular signaling as well. Our case involving elevated cytokines in COVID-19 elucidated how the approach can be used to capture known cytokine–cytokine interactions and use the data as a starting point for subsequent studies. The advantage and purpose of these cytokine-focused meta-edges is that they could enable the identification of previously unknown drug targets by focusing on upstream cytokines, especially if direct intervention on a downstream cytokine would be unavailable or not preferable. In addition, it could allow researchers to predict the potential ways an administered drug could alter downstream cytokine responses and the associated cell types involved in intercellular communication. However, our current model is limited by the assumption that all cytokines can reach all cell types carrying their respective receptors, and affect the release of other cytokines, while these criteria may not always be fulfilled in vivo [52]. Additionally, currently not all included interactions have been captured by the literature, and as such, future releases should iteratively improve upon the annotation of the interactions. In the future, modelling steps should also consider the involvement of intracellular signalling steps to better infer cytokine activity, and spatiotemporal data if available, to enhance precision when predicting putative cytokine–cytokine interactions and reduce the chance of false positives.

In conclusion, we have developed a novel network resource aimed at capturing the most prevalent cell–cell and cytokine–cytokine interactions in healthy individuals. Furthermore, this resource can be overlaid with disease-specific expression data from patients to reveal changes occurring in cytokine-mediated intercellular communication during disease states. Hence, CytokineLink can act as a powerful platform for hypothesis-generation i.e., for deconvoluting the complexities of cytokine signalling in inflammatory and infectious diseases to reveal potential cytokine drug targets, or for predicting the downstream effects of drugs on cytokine responses.

## Figures and Tables

**Figure 1 cells-10-02242-f001:**
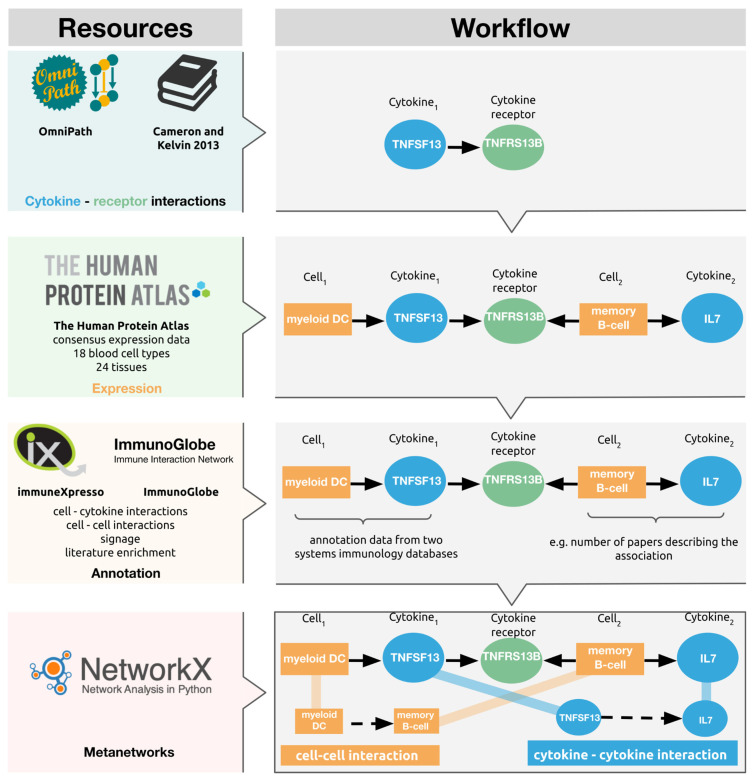
The construction steps and main sources of CytokineLink. Cytokine–receptor data was obtained from OmniPath and a relevant literature source. RNA enrichment data was downloaded from the Human Protein Atlas to assign cytokines and cytokine receptors to source tissue and cell types. The combination of cytokine–receptor interactions, and tissue/cell–cytokine/receptor interactions was used to generate base networks containing tissue–cytokine–receptor interactions, from which abstracted meta-edges were created. The cell–cytokine associations were further annotated using information from two systems of immunology databases, immuneXpresso and ImmunoGlobe. The cell–cell meta-edges show the intercellular interactions mediated by the individual cytokines. The cytokine–cytokine edges indicate the potential ways by which the production of a cytokine can alter the production of another, by binding to its receptor carried by a cell type expressing the secondary cytokine.

**Figure 2 cells-10-02242-f002:**
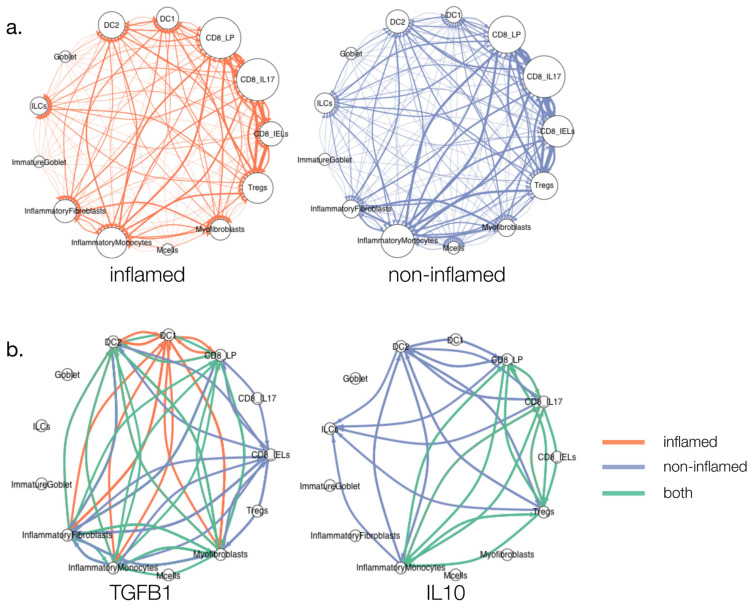
Cytokine-specific rewiring of intercellular interactions in ulcerative colitis. Global cell–cell interaction (**a**) and cytokine-specific cell–cell interaction (**b**) networks in inflamed and non-inflamed UC tissues. (**a**) Global cell–cell interaction networks showed only a few differences in the cell types interacting with each other between non-inflamed and inflamed UC states. More interactions between inflammatory fibroblasts and type 1 dendritic cells (DC1s) were present in inflamed UC compared to non-inflamed UC. The size of the nodes corresponds to the degree. (**b**) Cytokine-specific cell–cell interaction networks revealed more notable differences between non-inflamed and inflamed UC states, particularly with the cytokines IL10 and TFB1. IL10 was found to be produced by regulatory T cells (Tregs) in both inflamed and non-inflamed UC tissues, but in non-inflamed conditions, IL10 was found to interact with additional cell types including innate lymphoid cells (ILCs) and type 2 dendritic cells (DC2s). In inflamed UC, TGFB1 signalling shifted to interactions occurring mostly between DC1s and inflammatory fibroblasts, myofibroblasts, DC2s, and CD8 lamina propria (LP) T cells.

**Figure 3 cells-10-02242-f003:**
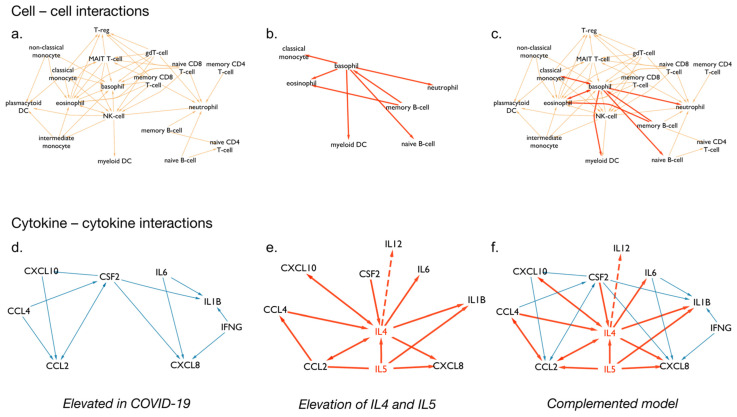
Metanetworks of cytokines increased in COVID-19 patients. (**a**) Intercellular interactions involving cytokines elevated in COVID-19 patients. (**b**) Intercellular interactions mediated by IL4 and IL5. (**c**) Combined intercellular network of cytokines elevated in COVID-19 patients, and intercellular interactions mediated by IL4 and IL5. (**d**) Cytokine–cytokine involving cytokines elevated in COVID-19 patients. (**e**) Cytokine–cytokine interactions involving IL4 and IL5. (**f**) Combined cytokine–cytokine network. The IL4-IL12 interaction is highlighted with dashed lines. IL4 is increased in severe COVID-19, potentially repressing IL12, whose levels are comparable to healthy controls in severe cases [37,38]. Mutual edges were collapsed, shown with arrows on both ends to reduce complexity.

**Table 1 cells-10-02242-t001:** Datasets used to investigate the reliability of cytokine–cytokine interactions. Gene expression datasets following cytokine inhibitor treatment were used to determine the significance of the cytokine–cytokine interactions captured by CytokineLink for a subset of cytokines.

Dataset	Reference	Drug	Inhibited Cytokine	*p*-Value
GSE16879	[13]	infliximab	TNFα	0.01504
GSE92415	(Li et al., 2018, unpublished data)	golimumab	TNFα	0.01525
GSE93777	[14]	tocilizumab	IL6	0.00587

**Table 2 cells-10-02242-t002:** Interaction data and annotations in CytokineLink. Annotated interactions refer to that at least one part of the underlying cell → cytokine interactions being listed in the literature in the ImmunoGlobe and immuneXpresso databases.

Tissues		24
Blood cell types		18
Cytokines		115
Cytokine–receptor pairs		260
Interactions		
cell–cell	All interactions between two cells regardless of receptor usage	581
	All interactions listed between two cells, mediated by different receptors	1118
cytokine–cytokine	All interactions between two cytokines regardless of receptor usage	2818
	All interactions listed between two cytokines, mediated by different receptors	9195
Annotated interactions		
cell–cell	Number of cell–cell interactions with annotated cell–cytokine relationships	74
	Percentage of cell–cell interactions with annotated cell–cytokine relationships	6.7%
cytokine–cytokine	Number of cytokine–cytokine interactions with annotated cell–cytokine relationships	1673
	Percentage of cytokine–cytokine interactions with annotated cell–cytokine relationships	18.2%

## Data Availability

The original contributions presented in the study are included in the article’s Appendix A and available at https://github.com/korcsmarosgroup/CytokineLink (accessed on 2 August 2021).

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
