# Peer review of "CytokineLink: A Cytokine Communication Map to Analyse Immune Responses—Case Studies in Inflammatory Bowel Disease and COVID-19"

_cells, 2021, doi:10.3390/cells10092242_

Round 1
Reviewer 1 Report
The proposed model of network is interesting.
The data is presented in an appropriate way. The text in the results add to the data and it is not repetitive. Statistically significant results are clear. It is clear which results are with practical meaning. Results are discussed from different angles and placed into context without being overinterpreted.
The conclusions answer the aim of the study. The conclusions are supported by references and own results.
Specific comments on weaknesses of the article and what could be improved:
Major points
- The title may be precised - taking into account that COVID-19 is the primary infection that is described within the paper.
- What are the clinical implications of these results? In the abstract it is mentioned - to develop and apply new therapies, i.e. anti-cytokine therapy. However, this should be exapnded in the discussion.
Minor points
- Please, state the limitations of the study
- Abstract should be revised, especially when mentioned COVID-19 as an example of infection where this cytokine network is important for the pathogenesis.
Author Response
We thank the reviewer for their insight, which helped us make improvements to the manuscript. We made the following changes to the text:
Major points:
- We changed the title to better reflect the illnesses used to highlight the capabilities of the model. The new title is: “CytokineLink: a cytokine communication map to analyse immune responses - case studies in inflammatory bowel disease and COVID-19”
- We have expanded upon the clinical implications of the results in the discussion section, for example highlighting the potential role the generated metanetworks could fulfill when designing drug intervention strategies.
Minor points:
- We have added the limitations of the study in the discussion section, discussing the assumptions underlying our model, and how they could be improved in future releases.
- We have modified the relevant segment of the abstract to more specifically describe the COVID-19 related findings.
Reviewer 2 Report
The authors have developed a novel network resource called CytokineLink, which shows widespread cell-cell and cytokine-cytokine interactions in healthy individuals. Furthermore, this resource can be used to reveal changes occurring in cytokine-mediated intercellular communication during disease states. Furthermore, it can be used to analyzing the complexities of cytokine signaling in different diseases, find potential cytokine drug targets, and effects of drugs or therapies on cytokine responses.
The thought and execution behind the study are good.
Author Response
We thank the Reviewer for their kind comments.